# *Arabidopsis* Trichome Contains Two Plasma Membrane Domains with Different Lipid Compositions Which Attract Distinct EXO70 Subunits

**DOI:** 10.3390/ijms20153803

**Published:** 2019-08-03

**Authors:** Zdeňka Kubátová, Přemysl Pejchar, Martin Potocký, Juraj Sekereš, Viktor Žárský, Ivan Kulich

**Affiliations:** 1Department of Experimental Plant Biology, Faculty of Science, Charles University, 12800 Prague, Czech Republic; 2Institute of Experimental Botany, Czech Academy of Sciences, 165 02 Prague, Czech Republic

**Keywords:** cell wall, EXO70, exocyst complex, phosphatidic acid, phosphatidylinositol 4,5-bisphosphate, phospholipids, plasma membrane domains, polar exocytosis, trichome

## Abstract

Plasma membrane (PM) lipid composition and domain organization are modulated by polarized exocytosis. Conversely, targeting of secretory vesicles at specific domains in the PM is carried out by exocyst complexes, which contain EXO70 subunits that play a significant role in the final recognition of the target membrane. As we have shown previously, a mature *Arabidopsis* trichome contains a basal domain with a thin cell wall and an apical domain with a thick secondary cell wall, which is developed in an EXO70H4-dependent manner. These domains are separated by a cell wall structure named the Ortmannian ring. Using phospholipid markers, we demonstrate that there are two distinct PM domains corresponding to these cell wall domains. The apical domain is enriched in phosphatidic acid (PA) and phosphatidylserine, with an undetectable amount of phosphatidylinositol 4,5-bisphosphate (PIP_2_), whereas the basal domain is PIP_2_-rich. While the apical domain recruits EXO70H4, the basal domain recruits EXO70A1, which corresponds to the lipid-binding capacities of these two paralogs. Loss of EXO70H4 results in a loss of the Ortmannian ring border and decreased apical PA accumulation, which causes the PA and PIP_2_ domains to merge together. Using transmission electron microscopy, we describe these accumulations as a unique anatomical feature of the apical cell wall—radially distributed rod-shaped membranous pockets, where both EXO70H4 and lipid markers are immobilized.

## 1. Introduction

Despite their unicellularity, *Arabidopsis* trichomes grow into extraordinarily shaped and precisely polarized structures, which makes them a potent model of cell polarization and morphogenesis [1]. According to [2], trichome development can be divided into six stages. During the first stage, several rounds of endoreduplication lead to an increased DNA content and, later, to a remarkably large trichome cell. The second and third stages are crucial for trichome shaping because they involve oriented growth above the surface and two or three branching events. During the fourth and fifth stages, the branches elongate significantly and the trichome reaches its final size and shape. When the growth is complete, the trichome enters the last stage of its development, which is maturation of the cell wall. In this stage, the trichome is divided into two markedly different domains—the basal domain, with a thin cell wall, and the apical domain, with an extremely thick cell wall. This latter secondary cell wall (SCW) consists of at least two distinct layers—the outer, cellulose-rich layer and the inner, callose-rich layer [3]. The inner layer is autofluorescent and silicified in a callose-dependent manner [4]. The apical cell wall domain also contains surface papillae, which accumulate cuticular waxes that may be different from those in other epidermal cells [5]. Basal and apical trichome domains are separated by a ring-shaped, callose-rich structure named the Ortmannian ring [3].

The development of this cell wall is dependent on the EXO70H4 exocyst subunit, which is necessary for callose synthase delivery to the plasma membrane (PM) [4]. EXO70H4 is a subunit of the exocyst complex, which is a eukaryotic protein complex involved in the tethering of post-Golgi secretory vesicles, which carry membrane and cell wall components to the PM [6]. The exocyst is composed of eight different subunits forming a functional complex, including Sec3, Sec5, Sec6, Sec8, Sec10, Sec15, Exo70, and Exo84 (for review, see [7,8]). Exo70 and Sec3 have a special position in the complex, as both of these subunits are able to directly bind to the target membrane lipids [9,10,11], thus regulating where the secretion will occur [12]. In yeast and mammals, the Exo70 subunit recruits the rest of the exocyst complex to the PM via a specific interaction of the EXO70 C-terminus with phosphatidylinositol 4,5-bisphosphate (PIP_2_) [9,13]. Sec3 is then responsible for initiating the binary SNARE (Soluble NSF Attachment Protein Receptor) complex [14]. The crosstalk of phospholipids with their protein partners and its importance in plant cell polarity determination and membrane traffic regulation was summarized in [15].

The *Arabidopsis* genome contains 23 EXO70 paralogs [16,17,18]. This multiplicity of EXO70 subunits in plants led to the hypothesis that EXO70s may have divergent lipid-binding properties and thus regulate exocytosis in distinct PM domains within a single cell [19]. So far, there is one documented example of such domain separation observed for NtEXO70A1 and NtEXO70B1 paralogs in *Nicotiana tabacum* pollen tubes, where EXO70A1 was more apically localized than EXO70B1 [20]. There is also apparent functional specialization among the EXO70 paralogs, as neither EXO70A1, B1, nor any other of 18 tested paralogs can complement the *exo70H4-1* loss-of-function mutant phenotype [4]. In this study, we document the differential localization of EXO70H4 and EXO70A1 in the mature trichome and show that EXO70H4 is required for the development and separation of two PM domains with different compositions of signaling phospholipids and distinct abilities to attract other EXO70 members.

## 2. Results

### 2.1. Mature Arabidopsis Trichomes Contain Two Distinct Lipid Domains and EXO70H4 is Involved in Their Formation

To investigate the distribution of the PM lipids in the *Arabidopsis* trichome, we observed localization of several fluorescent phospholipid markers. We focused on phosphatidylinositol-4-phosphate (PI4P) [21], phosphatidylinositol 4,5-bisphosphate (PIP_2_) [21], phosphatidic acid (PA) [22,23], and phosphatidylserine (PS) membrane lipid markers [23]. While the PIP_2_ marker was localized almost exclusively to the trichome base beneath the Ortmannian ring (Figure 1), the PI4P and PA markers were distributed evenly around the whole PM, with their signals enhanced at the Ortmannian ring and in intramural pockets throughout the apical domain (Figure 1). However, the PI4P marker signal inside these intramural pockets was weaker than the PA marker signal, which is highly accumulated there (Figure 1). The identities of these pockets will be described in more detail in Section 2.3. The PS marker was mostly visible at the apical membrane domain (Figure 1 and Appendix A). No signal was observed at the base, but a weak signal may have remained undetected due to a strong cytoplasmic background of this marker line. Thus, we concluded that a mature *Arabidopsis* trichome contains two distinct membrane domains with different lipid compositions.

Next, we wanted to investigate when these two domains differentiated during trichome development. For this, we used a double marker line expressing a PA marker tagged with mCitrine and a PIP_2_ marker tagged with mCherry (mCH), provided by the authors of [23]. In the wild type (WT) background, the distribution of the lipid markers was consistent with previous observations (Figure 2a). Distinct domains only appeared at stage 6 of trichome development, along with the establishment of the Ortmannian ring. In the WT trichomes of stage 4 (elongation), where the Ortmannian ring was not formed yet, PM domains were not visible (Figure 2b). The PIP_2_ marker was evenly distributed all around the PM and the PA marker was not bound to the membrane at all (Figure 2b), suggesting that PA is not present in young trichome PM. Later, we introduced both lipid markers into the *exo70H4-1* mutant background, which was unable to finish the cell wall maturation [3]. Here, the membrane domains lacked a sharp border and were poorly visible (Figure 2c). We concluded that the establishment of the apical and basal membrane domains occurred while the EXO70H4-dependent SCW layer in the apical domain was formed, which depended on the SCW formation.

### 2.2. Trichome Apical and Basal Plasma Membrane Domains Recruit Different EXO70 Proteins

To address the biological relevance of the trichome lipid domain distribution, we observed multiple EXO70 proteins under the control of the EXO70H4 promoter. EXO70A1 was previously found to bind to PIP_2_ [24] and to localize to the PIP_2_-rich region of the pollen tube [20]. Corresponding with its lipid affinity, EXO70A1 localized preferentially to the basal trichome domain and, in many cases, the apical trichome domain was completely devoid of EXO70A1, although there was a certain degree of variability and it was often seen also within the Ortmannian ring area (Figure 3a, Appendix A). As reported previously [4], a lack of EXO70H4 resulted in EXO70A1 localizing all around the trichome PM (Figure 3a). To verify that EXO70A1 is natively present in *Arabidopsis* trichomes, we also generated a reporter construct of the EXO70A1 promoter fused to 2xGFP (green fluorescent protein). This revealed that EXO70A1 was indeed expressed in the mature trichome (Figure 3b).

EXO70H4 is known to localize to the apical trichome domain and to the Ortmannian ring [4]. To correlate its localization together with EXO70A1, we co-transformed EXO70H4p::mCHERRY-EXO70H4 (mCH-EXO70H4) with EXO70H4p::GFP-EXO70A1 (GFP-EXO70A1) In this case, the EXO70 isoforms localized to distinct and mostly non-overlapping PM domains that corresponded to the apical and basal trichome domains (Figure 3c).

We also investigated the localization of the remaining EXO70s, which we observed previously in the *exo70H4-1* mutant background [4]. This time, we expressed the constructs in a WT background to see their localization when it was unaffected by the *exo70H4-1* mutation. Eight paralogs were tested—EXO70A1, A2, B1, C1, D2, F1, H7, and H8. The only paralogs found to localize to the PM in the WT background in addition to EXO70A1 were EXO70A2 (localized similarly to EXO70A1) and EXO70H8. EXO70H8 strongly accumulated at the PA-rich domain in the apical part of the trichome, including Ortmannian ring and cell wall ingrowths. Although EXO70H8 mimicked EXO70H4 localization in the WT trichome, it was not capable of functionally complementing the *exo70H4-1* mutant phenotype in our previous cross-complementation study [4]. The other EXO70 paralogs that were tested remained in the cytoplasm or nucleus, as in the previous experiments in the *exo70H4-1* background.

Since the lipid-binding capacities of EXO70H4 have not yet been described, we performed a protein–lipid overlay assay with in vitro translated HA-tagged EXO70H4. This revealed a clear affinity to PS and conceivably PA, but no apparent binding to PIP_2_ (Figure 3d). This corresponded well with the colocalization of EXO70H4 with PA and PS markers in the apical domain of the mature trichome. Based on these and previously published data, we concluded that different EXO70 proteins exhibit a specific capacity to bind membrane lipids and thus are recruited to distinct PM domains, contributing to the biogenesis of different cell wall domains within a single plant cell.

### 2.3. The Apical Cell Wall Contains Entrapped Membranous Pockets

While observing the apical trichome domain, we noticed that a large portion of the signal came from within the cell wall. This was true for all of the constructs which localized to the apical membrane domain, including markers of PA, PS, PI4P, mCh-EXO70H4, GFP-EXO70H8, and occasionally also EXO70A1 and EXO70A2 with a weak signal. This strange intramural localization directed our further investigation. The signal within the SCW was distributed in a pattern remarkably similar to the callose deposits shown before [3,4]. The radially distributed rays of signal were visibly embedded within the cell wall and were well-apparent on fluorescence microscopy optical cuts of matured branches together with a wrinkled PM–SCW interface (Figure 4a,b). To be sure that this was indeed the fluorophore signal and not cell wall autofluorescence, we performed lambda scans in plants expressing GFP-EXO70H4, mCh-EXO70H4, YFP-PA marker, and untransformed control. In all of these cases, the expected emission peak was observed (Appendix A), with no such peaks in the negative control. To check if these intramural signals were still connected to the rest of the cytoplasm, we performed a fluorescence recovery after photobleaching (FRAP) experiment on the PI4P marker, which revealed that there was no detectable recovery of the signal, unlike in the case of the signal bleached within the apparent continuum of the PM (Appendix A). We therefore hypothesized that the intramural signal may come from the PM and cell interior, being physically entrapped within the cell wall pockets.

To investigate the SCW structure in more detail, we used transmission electron microscopy (TEM). This revealed internal cell wall structures, supporting our observations of oblong traversing pockets with light microscopy. TEM cross-sections of mature WT trichome branches displayed SCW-transpassing structures, probably aggregates of entrapped secretory lipid membranes, proteins, and the cell interior, often arranged in a radial pattern of concentric transpassing channels (Figure 4c, left). These intramural pockets were obviously separated from the continuum of cytoplasm (Figure 4c, middle), but their PM origin was apparent from images where pieces of electron-dense, non-cell wall materials were embedded into the cell wall (Figure 4c, right).

## 3. Discussion

We showed that the mature *Arabidopsis* trichome contains, along with two cell wall domains, two distinct PM domains that differ in their phospholipid composition and also in their ability to recruit different EXO70 proteins. EXO70A1 is recruited to the basal trichome domain, which is PIP_2_-rich and contains a thin, pectin-rich cell wall [25]. EXO70H4 is recruited to the apical domain, which is PS- and PA-rich and has a thick, autofluorescent cell wall [3].

There are a number of different examples of cell types having different phospholipid domains and many of them can be linked with differential cell wall deposition. Pollen grains also accumulate PIP_2_ at the site of the future aperture, which is an area marked with a thin cell wall [26]. Similarly, the pollen tube accumulates PIP_2_ at its tip, while PS and PA are in the shank [22,23]. This corresponds with the apical cell wall of the pollen tube being almost solely made of secreted pectins, while callose and cellulose are deposited in the shank [27]. This domain separation in the pollen tube is further marked by differential membrane localization of EXO70A1 and EXO70B1 in tobacco [20]. In contrast to the pollen tube, trichomes are extremely large and static structures. The border of the two membrane domains appears to be very sharp and defined by the Ortmannian ring. A lack of a sharp border between these two domains and a partial loss of polarity in the *exo70H4-1* mutant suggests that there is a positive feedback loop and that EXO70s have a role in domain development.

The observed lack of PIP_2_ and abundance of PA in the apical domain could also imply that, in the apical domain, PIP_2_ is metabolized to PA. In general, PA is formed through two pathways: First, by the direct hydrolysis of structural phospholipids by phospholipase D and second, through the consecutive actions of phospholipase C and diacylglycerol kinase, where the hydrolysis of PIP_2_ produces inositol-1,4,5-trisphosphate (released into the cytosol) and diacylglycerol within the PM, which is quickly phosphorylated at the membrane into PA by diacylglycerol kinase. PA produced by phospholipase Dα1 is a crucial signaling lipid, mediating the abscisic acid response in guard cells, where its role in NADPH oxidase-mediated reactive oxygen species (ROS) production was clearly demonstrated [28]. Both PA and phospholipase Dα1 have previously been linked with ROS production [29]. NADPH oxidase-mediated ROS production acts in the plant defense response [30,31]. In plants, PA is also produced in response to several stress factors, including pathogen attack [32]. EXO70H4 is also induced by *flagellin 22* in epidermal pavement cells [4]. The apical cell wall domain of the trichome is also remarkably similar to the defense papillae, as they both accumulate callose deposits, phenolic compounds [33], and an extracellular signal of membrane proteins (such as SYP121—SYNTAXIN OF PLANTS 121 [34]). We therefore suggest that the mechanism and domain organization in the trichome may be a manifestation of a general response to pathogen attack, which runs constitutively in the trichome.

It was shown that binding of PA to its protein interactors is enhanced by a negative curvature stress and that a complex membrane lipid composition strongly influences lipid–protein interactions [35]. Surprisingly, EXO70 by itself is able to induce negative membrane curvature, specifically through the homodimerization mechanism, as was demonstrated in mammals [36]. Together, these biophysical properties could induce PM deformation sufficient to form extreme membrane curvatures, leading to the formation of membranous pockets within the SCW. Another factor contributing to the pocket formation may be the cuticular wax migration across the cell wall. The apical trichome domain displays several unique features, including surface papillae formation. Papillae start to form at stage 5 of trichome development (expansion) and continue developing during stage 6 (maturation). At these stages, the apical cell wall is already quite thick (>2 µm) [37]. Papillae are little bumps filled with lipophilic cuticular substances. These waxes differ from those of other epidermal cells by a high content of C35+ alkanes [5]. How these compounds get through the thick cell wall to the surface is not understood. Membranous pockets may drift along the migration routes of the cuticular waxes through the inner layer of the SCW. This feature may be very useful for the study of cuticular wax migration across the cell wall. Another intriguing possibility is that formation of these membranous pockets within the SCW is directly linked to the mechanism of callose deposition during SCW biogenesis—a possibility we aim to study.

## 4. Materials and Methods

### 4.1. Plant Material

Plants were grown in Jiffy soil pellets in standard growth chamber conditions (long day 16 h:8 h, 100–120 µM photosynthetically active radiation m^−2^ s^−1^). The *exo70H4-1* mutant line was described previously [3]. As WT control, outcrossed WT plants were used. Seeds of PI4P and PIP_2_ markers were obtained from Yvon Jaillais and are described in detail in [21]. From this set, we used line P21Y (mCitrine-2xPH FAPP PI4P binding domain), line P24Y (mCitrine-2xPH^PLC^ PIP_2_ binding domain), and line P24R (mCherry-2xPH^PLC^ PIP_2_ binding domain). Two different PA markers were used in this study. In Figure 1, it was YFP:NES-2xSpo20p (cloning described below) and in Figure 2 and Appendix A, it was a double marker line expressing a PA marker tagged with mCitrine (PAY) and a PIP_2_ marker tagged with mCherry (P24R, mentioned above), provided by [23]. PS marker cloning is described below.

### 4.2. Confocal Microscopy

Confocal microscopy images were taken on Zeiss LSM880 with C-Apochromat 403/1.2 W Korr FCS M27 objective [GFP (488): 508–540 nm, chlorophyll (488) 650–721 nm, cell wall autofluorescence (405) 426–502, mCherry (561) 597–641]. Images were processed using the Fiji platform [38].

### 4.3. Trichome Isolation and TEM

Leaves of 4-week-old plants were collected and incubated for 3 h in falcon tubes in a solution of acetic acid:ethanol (1:3) and then washed three times with deionized water. Washed leaves were transferred to a solution of 150 mM KH_2_PO_4_ pH 9.5 and incubated overnight with shaking at 150–180 rpm. Released trichomes were collected by centrifugation (1 min, 1000 G, no break). For TEM, isolated trichomes were fixed for 24 h in 2.5% (*v*/*v*) glutaraldehyde in 0.1 M cacodylate buffer (pH 7.2) at 4 °C and postfixed in 2% (*w*/*v*) OsO4 in the same buffer. Fixed samples were dehydrated through an ascending ethanol and acetone series and embedded in Epon–Araldite.

### 4.4. Cloning and Stable Transgenic Line Preparation

For cloning of the expression reporter construct EXO70A1p::GFP:GFP, the multisite gateway approach was used. The EXO70A1 promoter (1 kb upstream, EXO70A1 prom for and EXO70A1 prom rev listed in Appendix A were used) was subcloned into pDONORP4-P1r and sequenced using M13 primers. GFP constructs—pEN-L1-F-L2 and pEN-R2-GFP-L3,0—and destination vector pB7m34GW were obtained from [39]. An expression clone was assembled from these by a multisite reaction using Gateway LR clonase (Thermo Fischer, Waltham, MA, USA). All EXO70H4p::GFP-EXO70XY constructs used in this study were previously described in [4].

To prepare the construct for the PA marker (pUBQ::YFP:NES-2xSpo20p PABD), first NES-2xSpo20p was amplified from YFP:Spo20p-PABD [22] and cloned together with the ubiquitin (UBQ) promoter and YFP into the binary vector pHD71 (kindly provided by Dr. Benedikt Kost).

To prepare the construct for the PS marker (pUBQ::YFP:C2^LACT^), YFP-C2^LACT^ from Lat52:YFP-C2^LACT^ [22] was cloned together with the UBQ promoter into the binary vector pHD71; the primers used are listed in Appendix A.

Final constructs were electroporated into competent cells of *Agrobacterium tumefaciens*, strain GV3101. Col-0 wild type plants were transformed by the floral dip method [40] and transformants were selected on kanamycin plates or soil by BASTA spraying (150 mg/L of glufosinate-ammonium). No fewer than five individual transformants were observed in each experiment and at least two biological replicates were made.

### 4.5. Protein–Lipid Overlay Assay

The protein–lipid overlay assay was performed using N-terminally HA-tagged EXO70H4 and echelon lipid strips. First, the EXO70H4 coding sequence was cloned into a pTNT vector (Promega, Fitchburg, WI, USA), optimized for in vitro coupled transcription and translation, using primers listed in Appendix A. The cloning was performed in two steps. First, the EXO70H4 coding sequence was amplified with the use of the primers EXO70H4-forward and EXO70H4-reverse. Subsequently, a megaprimer containing the HA-tag sequence and Kozak consensus sequence facilitating efficient translation initiation was used together with EXO70H4-reverse primer. The product was cloned into the pTNT vector via SalI and NotI restriction sites.

The construct was used as a template for in vitro coupled transcription and translation reactions using TNT^®^ SP6 High-Yield Wheat Germ Protein Expression System (Promega), according to the manufacturer’s instructions, in a total volume of 50 μL. To verify protein expression and stability, 10 μL of the yield was used in Western blotting. Since the protein was tagged on the N-terminus, Western blot analysis would have detected products of prematurely terminated translation, but none were observed (see Appendix A).

The remaining 40 μL of the reaction product was used for the protein–lipid overlay assay. The echelon lipid strip (Elcheon Biosciences, Salt Lake City, UT, USA) was blocked for one hour in blocking solution (50 mM Tris, 150 mM NaCl, 0.05% Tween 20, pH 7.6 + 3% bovine serum albumine). After three washes in wash solution (50 mM Tris, 150 mM NaCl, 0.05% Tween 20, pH 7.6), the strip was incubated for two hours with primary mouse anti-HA antibody (Sigma-Aldrich, St. Louis, MO, USA) diluted 1:1000 in blocking solution. After three washes with wash solution, the strip was subsequently incubated with secondary anti-mouse antibody conjugated to horseradish peroxidase (Promega), diluted 1:20,000 in blocking solution. Finally, the strip was washed three times with wash solution and incubated for three minutes with Amersham ECL Prime Western Blotting Detection Reagent (GE Healthcare, Chicago, IL, USA) and imaged with Bio-Rad Laboratories (Hercules, CA, USA) Chemidoc gel imaging system. The experiment was repeated twice with consistent results.

## Figures and Tables

**Figure 1 ijms-20-03803-f001:**
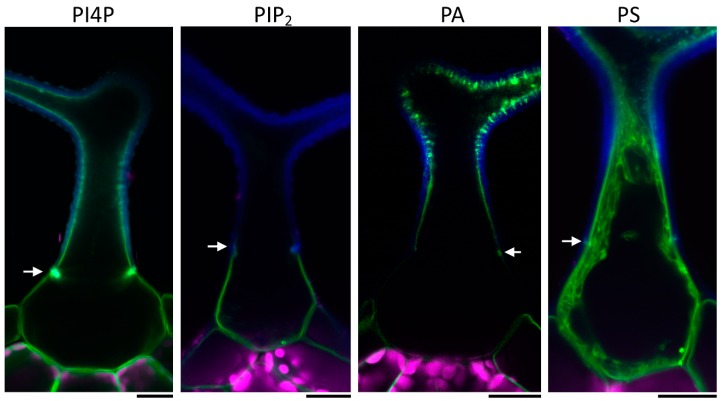
Representative images of different phospholipid markers in wild type mature trichome. PI4P—phosphatidylinositol-4-phosphate; PIP_2_—phosphatidylinositol 4,5-bisphosphate; PA—phosphatidic acid; PS—phosphatidylserine. Blue—cell wall autofluorescence; magenta—chlorophyll autofluorescence; green—mCitrine or YFP (yellow fluorescent protein). White arrows point at the Ortmannian ring. Scale bars = 20 µm.

**Figure 2 ijms-20-03803-f002:**
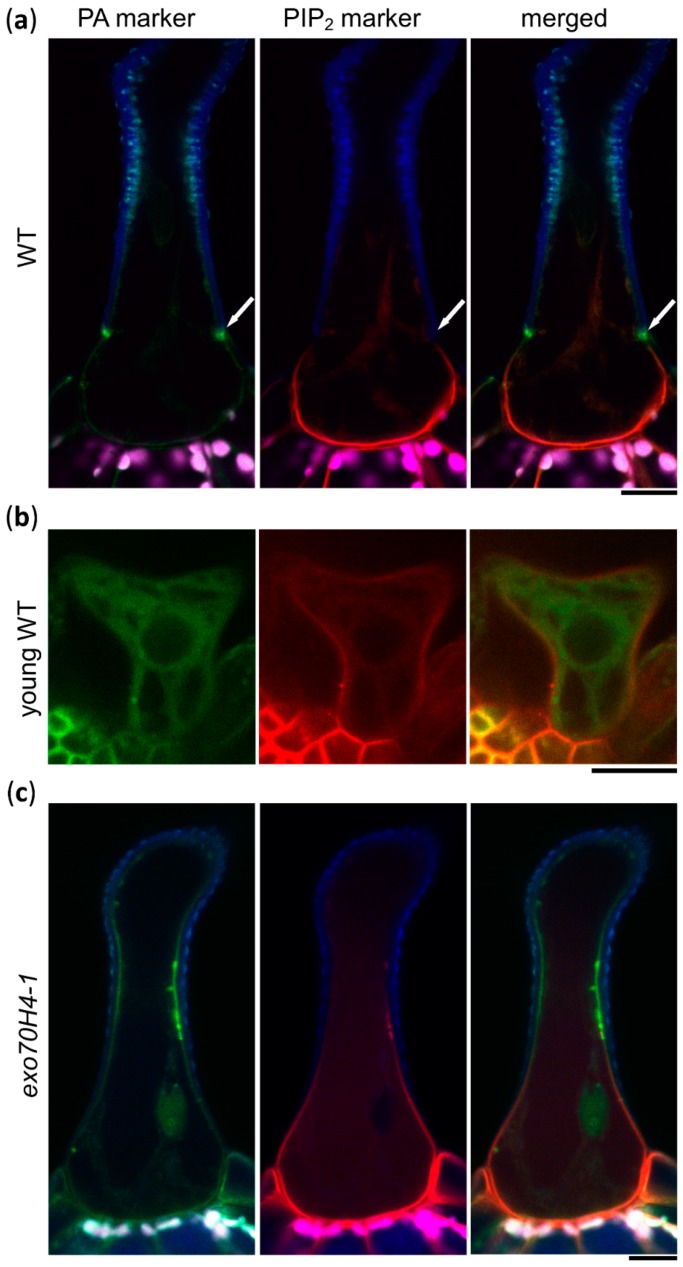
Lipid markers in WT and *exo70H4-1* trichomes. (**a**) Colocalization of PA and PIP_2_ markers in WT trichome; (**b**) colocalization of PA and PIP_2_ markers in WT trichome in the elongation stage; (**c**) colocalization of PA and PIP_2_ markers in an *exo70H4-1* mutant background. Blue—cell wall autofluorescence; magenta—chlorophyll autofluorescence; green—YFP; red—mCherry. White arrows point at the Ortmannian ring. Scale bars = 20 µm.

**Figure 3 ijms-20-03803-f003:**
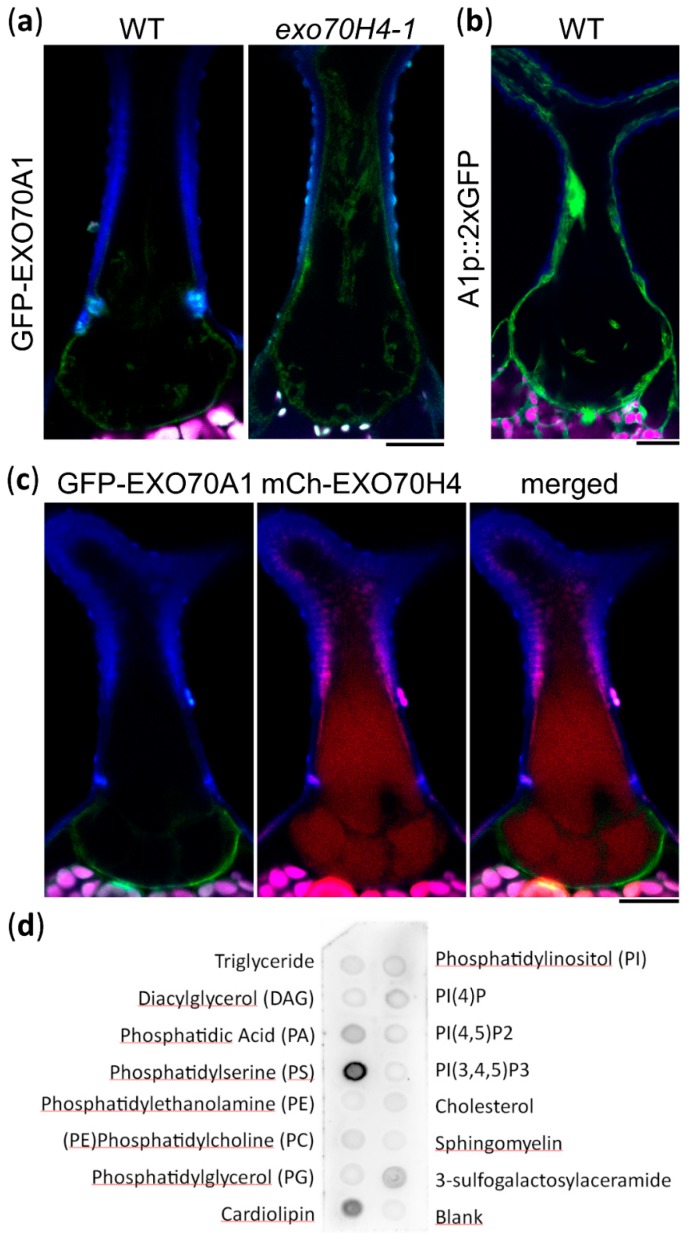
Trichome plasma membrane domains recruit different EXO70 proteins. (**a**) EXO70H4p::GFP-EXO70A1 (GFP-EXO70A1) preferentially localizes to the basal trichome domain in WT. This preference is lost in the *exo70H4-1* mutant. (**b**) EXO70A1p::GFP:GFP (A1p::2xGFP) expression marker in a WT trichome. (**c**) Colocalization of GFP-EXO70A1 with EXO70H4p::mCherry-EXO70H4 (mCh-EXO70H4). (**d**) Protein–lipid overlay assay of EXO70H4. Blue—cell wall autofluorescence; magenta—chlorophyll autofluorescence; green—GFP; red—mCherry. Scale bars = 20 µm.

**Figure 4 ijms-20-03803-f004:**
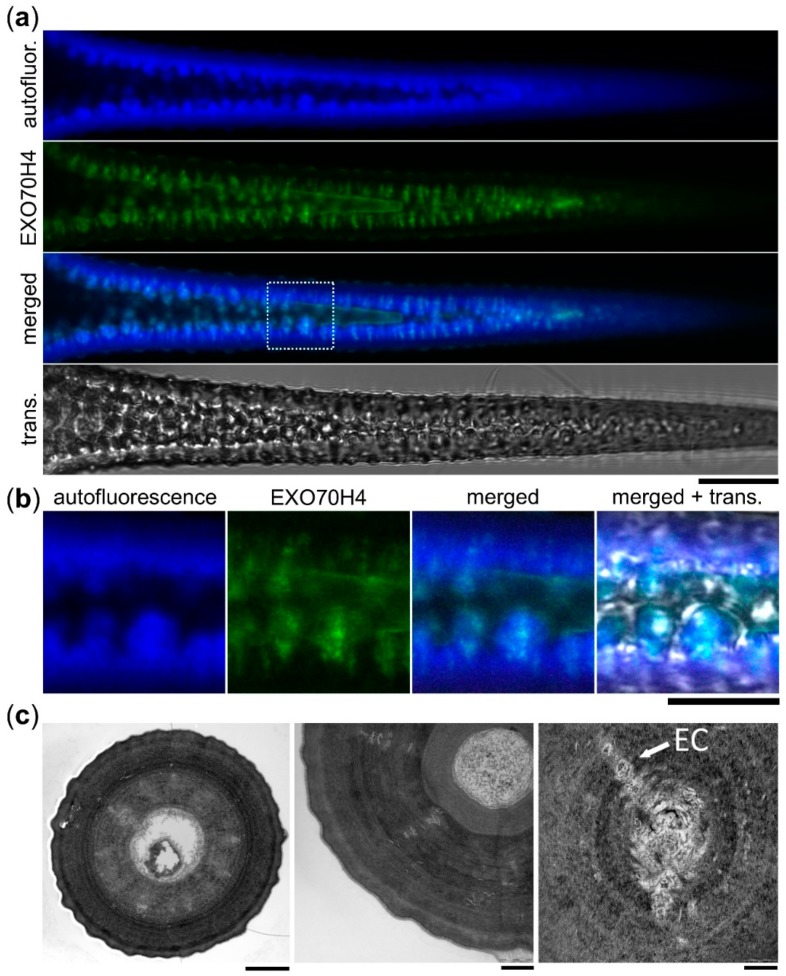
Plasma membrane proteins reside within the trichome apical cell wall. (**a**) GFP-EXO70H4 (EXO70H4) signal, cell wall autofluorescence (autofluor.), and transmission channel (trans.) at the branch of a mature WT trichome. The dotted square represents area enlarged in (**b**). Scale bar = 20 µm. (**b**) Detail of (**a**). Scale bar = 10 µm. (**c**) Left—TEM image of a cross-section of a trichome branch. Entrapped cell interior (EC) is visible as concentric rays. Scale bar = 3 µm. Middle—detail of entrapped cell interior obviously separated from the cytoplasm. Scale bar = 1 µm. Right—a detail of one concentric ray. Scale bar = 200 nm.

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
