# Peer review of "Arabidopsis Trichome Contains Two Plasma Membrane Domains with Different Lipid Compositions Which Attract Distinct EXO70 Subunits"

_ijms, 2019, doi:10.3390/ijms20153803_

Round 1
Reviewer 1 Report
In their manuscript, Kubátová et al. present a nice example of plant cells forming distinct cellular domains. They provide evidence that mature trichomes in Arabidopsis develop apical and basal domains that differ in the composition of plasma membrane lipids, attract different isoforms of the exocyst subunit Exo70, and have differences in the associated cell walls. The authors further show that the domain formation in trichomes is dependent on the presence of functional EXO70H4 and that the apical domains of trichomes have unusual, radially-distributed membrane pockets. The authors speculate that these pockets may be involved in biosynthesis of the thick cell wall associated with the apical domain.
The story is interesting and important; however, the presentation of the findings can be improved.
1. I am a bit concerned about the interpretation of the identity of membranous pockets. The authors seem to assume that these pockets are completely separated from the rest of the cell by the ingrowth of cell wall, but I do not think that the presented evidence strongly supports the conclusion that these pockets are entirely separate and do not have any contact with the rest of the cell. The TEM images shown in Fig. 4C do not provide a very clear view of these structures. I wonder if showing TEM of longitudinal sections of trichomes would allow to better see the outlines of the plasma membrane and membranous outgrowths.
2. The FRAP experiments that led authors to conclude that pockets are separate from the rest of the cell are also not very convincing. In particular, it is unclear why they were done on the line expressing the PI4P marker since PI4P is not specific to the apical domain (and according to Fig. 1 is not particularly prominent in the apical domain and does not show signal in radial rays) rather than on the line expressing the apical marker of PA? While the signal seems to be restored much faster in the general membrane than in the pockets, is it completely not restored in the pockets or does it eventually come back?
3. Lambda scans in Fig. S2: Since their purpose is to show the nature of fluorescent signal in the apical area with the rays of signals, they should all be performed on regions from this particular area of trichomes. Right now, some signals are collected from the general PM area (Fig.S2b) or even from the Ortmannian ring (Fig. S2d). Also, it is unclear what different colors shown in Fig. S2a-d indicate. Is there a reason to show these images in different colors? Is the goal here to simply indicate the areas for which lambda scans were obtained, in which case it can be shown in grey-scale, or is the color meaningful (if so, it has to be explained)?
4. Some of the images presented in Fig. 1 do not correspond well to the text that describes them. For example, the authors state that PI4P was evenly distributed around the whole PM, however, the image shown in Fig. 1 does not show much signal through a large portion of the apical domain. Similarly, the authors write about the PS marker that it is most visible at the apical membrane domain. Yet, given the extensive cytoplasmic location of this marker which obscures PM signal, it is really difficult to make conclusions about whether this marker indeed shows specific enrichment at the apical PM compared to the basal PM.
5. The ray-like patterns of signal that are obvious in the apical domain with several markers have to be introduced much earlier. Right now, although they are clearly visible already in Fig. 1-PA (corresponding text is in line 77), they are mentioned for the first time only in line 129 and without any explanation (“cell wall ingrowths”), which comes as a surprise to a reader – what are those? I would suggest that these ray-like patterns should be already mentioned during the description of signals produced by the PA marker (Fig. 1) and Exo70H4 (Fig. 2).
6. The second paragraph of the Results opens up with the goal of investigating the developmental timing of domain formation. Yet, instead of presenting the results of investigation of timing, the authors suddenly jump to WT vs. exo70H4-1 results. The timing, instead, is given short shrift. It would be better to first present the situation in young trichomes (and show images of the young trichomes at the top of the corresponding figure), then describe that the separation into domains occurs only after formation of OR, and only after that move onto the exo70h4-1 mutant experiments.
7. The relative intensity in the graphs in Fig. S2 and S3 – what is it relative to? Also, why indicate “16 bit”? Also, for the graph in Fig. S3 it would make more sense to label the units of the X-axis directly on the graph rather than mention it in the figure legend.
8. Please be consistent in the usage of terms. If the trichome development is divided into six stages (line 32), then continue calling them “stages”. Do not suddenly change to the term “phase” in the sentences that follow. Do not use the term “plasmalemma” only in Fig. S3 and its legend if the term “plasma membrane” is used throughout the manuscript.
9. Unclear what “vice versa” in line 12 refers to.
10. Unclear what “these” in line 164 refer to.
Reviewer 2 Report
The manuscript ijms 543215 entitled "Arabidopsis trichome contains two plasma membrane domains with different lipid composition which attract distinct EXO70 subunits" described that in Arabidopsis thaliana, the mature trichome contains two distinct plasma membrane domains which have different abilities to recruit EXO70 proteins, as well as they, differ in PL composition.
This is an important step forward as so far there are limited reports, to be precise only one in N. tabacum pollen tubes. The report is well prepared and scientifically sound. This report will lead to further enhancement of knowledge with the fact that EXO70 proteins have not strict lipid-binding properties and may regulate exocytosis in distinct plasma membrane domains within a single cell.
Author Response
No questions or comments were raised.